# Salvage vs. Primary Total Laryngectomy in Patients with Locally Advanced Laryngeal or Hypopharyngeal Carcinoma: Oncologic Outcomes and Their Predictive Factors

**DOI:** 10.3390/jcm12041305

**Published:** 2023-02-07

**Authors:** Shahin Tahan Shoushtari, Jocelyn Gal, Emmanuel Chamorey, Renaud Schiappa, Olivier Dassonville, Gilles Poissonnet, Déborah Aloi, Médéric Barret, Inga Safta, Esma Saada, Anne Sudaka, Dorian Culié, Alexandre Bozec

**Affiliations:** 1Institut Universitaire de la Face et du Cou, Centre Antoine Lacassagne, 33 Avenue de Valombrose, 06189 Nice, France; 2Department of Statistics, Centre Antoine Lacassagne, 33 Avenue de Valombrose, 06189 Nice, France; 3Department of Radiation Oncology, Centre Antoine Lacassagne, 33 Avenue de Valombrose, 06189 Nice, France; 4Department of Medical Oncology, Centre Antoine Lacassagne, 33 Avenue de Valombrose, 06189 Nice, France; 5Department of Pathology, Centre Antoine Lacassagne, 33 Avenue de Valombrose, 06189 Nice, France; 6Faculty of medicine, Côte d’Azur University, 06107 Nice, France

**Keywords:** laryngeal carcinoma, hypopharyngeal carcinoma, larynx preservation, total laryngectomy, salvage surgery

## Abstract

Background: The aims of this study were to compare the survival outcomes of salvage vs. primary total laryngectomy (TL) in patients with locally advanced laryngeal or hypopharyngeal carcinoma and to determine their predictive factors. Methods: Overall (OS), cause-specific (CSS) and recurrence-free survival (RFS) of primary vs. salvage TL were compared in univariate and multivariate analysis taking into account other potential predictive factors (tumor site, tumor stage, comorbidity level etc.). Results: A total of 234 patients were included in this study. Five-year OS was 53% and 25% for the primary and salvage TL groups, respectively. Multivariate analysis confirmed the independent negative impact of salvage TL on OS (*p* = 0.0008), CSS (*p* < 0.0001) and RFS (*p* < 0.0001). Hypopharyngeal tumor site, ASA score ≥ 3, N-stage ≥ 2a and positive surgical margins were the main other predictors of oncologic outcomes. Conclusions: Salvage TL is associated with significantly worse survival rates than primary TL highlighting the need for careful selection of patients who are candidates for larynx preservation. The predictive factors of survival outcomes identified here should be considered in the therapeutic decision-making, especially in the setting of salvage TL, given the poor prognosis of these patients.

## 1. Introduction

There are different validated therapeutic options to preserve laryngeal functions in patients with laryngeal or hypopharyngeal carcinoma [1,2,3]. Patients with early stage disease are offered definitive radiation therapy (RT) or partial surgery (i.e., endoscopic laser CO_2_ surgery, transoral robotic surgery or open partial pharyngo-laryngectomy) [1,2,3]. Since the 1990s and the development of laryngeal preservation (LP) protocols, most patients with locally advanced laryngeal and hypopharyngeal carcinoma are included in these complex therapeutic programs, combining either induction or concurrent chemotherapy with RT [3,4,5,6,7]. Gradually, radical surgery with total laryngectomy (TL) was therefore relegated to the salvage treatment of recurrent tumors [8,9].

The first LP studies showed that induction chemotherapy (ICT) with cisplatin and 5-fluorouracil (PF) followed by RT in good responders could preserve nearly 60% of larynx without deleterious impact on survival compared with the gold standard TL followed by RT [3,4,5]. Thereafter, other LP protocols have been developed investigating the role of definitive concurrent chemoradiation therapy (CRT) or of the combination of docetaxel (T) with the standard PF ICT regimen for LP [6,7]. Indeed, to date, there are two main validated modalities of LP in patients with locally advanced laryngeal or hypopharyngeal carcinoma: TPF ICT followed by RT in responders (preferred option in France and Europe) or definitive concurrent CRT (preferred option in North America) [3].

Twenty years after the development of LP, an epidemiological study in the USA covering 158,426 cases of laryngeal carcinoma from the “National Cancer Data Base” observed a decrease in the survival rate of patients since the mid-1990s, potentially in relationship with changes in therapeutic strategies and, in particular, the increase in the initial use of non-surgical treatments [10]. In this context, several studies have recently focused on the results of salvage TL but included a significant proportion of patients with initial early stage tumors and/or treated outside of a LP program [11,12,13,14,15,16,17,18,19]. As a result, it is largely unknown whether TL performed as a salvage procedure after failure of a LP program is associated with worse survival compared with primary TL for patients with locally advanced laryngeal or hypopharyngeal carcinoma.

The aims of this study were therefore to compare the survival outcomes of salvage TL with those of primary TL in patients with locally advanced laryngeal or hypopharyngeal carcinoma and to determine the predictive factors of oncologic results.

## 2. Materials and Methods

### 2.1. Ethical Considerations

The study protocol was reviewed and approved by the institutional ethics committee prior to the start of the study (approval code: F20220422105156). Informed consent was obtained from each of the participants.

### 2.2. Subjects

In this retrospective study, we included all patients who underwent TL between 2000 and 2020, at our institution, for a locally advanced (T3 or T4) squamous cell carcinoma of the larynx or hypopharynx, either as a primary therapeutic option or as a salvage procedure after failure of an LP program. The therapeutic strategy was elaborated for each specific patient during a multidisciplinary tumor board (MTB) discussion. The exclusion criteria were as follows: histology other than squamous cell carcinoma, tumor stage T1 or T2 at diagnosis, TL performed for an indication other than progressive cancer, salvage TL performed after a treatment other than a LP protocol (open or transoral partial laryngectomy, radiotherapy alone). LP protocols consisted of either cisplatin-based concurrent CRT or ICT (with PF before 2005 or TPF since 2005) followed by RT in good responders.

### 2.3. Follow-Up

Post-therapeutic follow-up was conducted in accordance with national guidelines. Patient clinical examination was scheduled every two months during the first two years, then every four months. A head and neck and thoracic CT-scan was performed 3 to 4 months after the initial treatment and once a year for the next 5 years, or in cases of clinical suspicion of tumor recurrence.

### 2.4. Main Outcome Measures

Patients’ general health status and comorbidity level were assessed using the American Society of Anesthesiologists (ASA) score. Tumor stage was determined according to the 8th edition (2017) of the Tumor Node Metastasis (TNM) classification of the Union for International Cancer Control and the American Joint Committee on Cancer.

Overall survival (OS), cause-specific survival (CSS) and recurrence-free survival (RFS) were determined by Kaplan–Meier analysis.

### 2.5. Statistical Analyses

Patients were divided into two groups based on the type of surgery: primary TL group and salvage TL group. Each patient’s clinical characteristics (gender, age, tumor site, T- and N-stage, etc.) of these two groups of patients were compared using Chi-square tests.

We analyzed the impact on OS, CSS and RFS of the type of surgery (primary vs. salvage TL) and of the following factors: age (< vs. >70 years), gender (male vs. female), ASA score (< vs. ≥3), primary tumor site (larynx vs. hypopharynx), preoperative tracheostomy, surgical margins (negative vs. positive), T-stage (3 vs. 4) and N-stage (< vs. ≥2A). Univariate analyses were performed using Log Rank tests. For multivariate analysis (conducted only when more than one factor was significant in univariate analyses), all variables associated with *p* < 0.05 in univariate analysis were included in Cox regression models with forward stepwise selection. The predictive ability of each Cox regression model was determined using the area under the receiver operating characteristic (ROC) curve (AUC) with a clinical endpoint fixed at 5 years.

All statistical analyses were performed at 5% alpha risk or 95% confidence interval by the biostatistician using R.3.0.1 software on Windows.

## 3. Results

### 3.1. Sample Description

Between 2000 and 2020, 353 patients underwent TL at our institution. After applying exclusion criteria (Figure 1), a total of 234 patients were included in the study (209 men and 25 women, mean age = 67.1 ± 10.5 years), 184 (79%) in the primary TL group and 50 (21%) in the salvage TL group. In the salvage TL group, 8 patients had received cisplatin-based concurrent CRT and 42 patients ICT with PF (n = 5) or TPF (n = 37) followed by RT.

Patients’ clinical characteristics for the whole cohort and according to the type of surgery (primary vs. salvage TL) are presented in Table 1. Compared with the primary TL group, patients of the salvage TL group displayed a significantly lower T-stage, a higher N-stage and higher rates of hypopharyngeal tumor, TL with circular pharyngectomy, pedicled or free flap pharyngeal reconstruction and involved surgical margins.

### 3.2. Oncologic Outcomes in the Primary vs. Salvage TL Groups

Median follow-up was 46.5 months (95% CI (95% confidence interval): 33.1–87.5 months). As shown in Figure 2a–c, OS, CSS and RFS were significantly lower (*p* < 0.001) for patients of the salvage TL group than for those of the primary TL group. Five-year OS, CSS and RFS were 25 vs. 53%, 34 vs. 64% and 16 vs. 45% for the salvage and primary TL groups, respectively.

### 3.3. Impact of Tumor Site on Oncologic Outcomes

Considering the observed differences in baseline patients’ clinical characteristics between the two treatment groups (primary vs. salvage TL) regarding the distribution of primary tumor site (larynx vs. hypopharynx), we compared the oncologic outcomes of primary vs. salvage TL separately, for patients with laryngeal carcinoma and for those with hypopharyngeal carcinoma. These results are shown in Table 2.

For patients with laryngeal carcinoma, primary TL was significantly associated with better OS (*p* = 0.002), CSS (*p* < 0.001) and RFS (*p* = 0.02) than salvage TL. For patients with hypopharyngeal carcinoma, primary TL tended to be associated with better OS (*p* = 0.06) and CSS (*p* = 0.1) than salvage TL. No significant difference was observed for RFS (*p* = 0.2).

Kaplan–Meier survival curves of the four subgroups of patients (1: larynx—primary TL, 2: larynx—salvage TL, 3: hypopharynx—primary TL, 4: hypopharynx—salvage TL) for OS, CSS and RFS are presented in Figure 3a–c.

### 3.4. Multivariate Analysis of the Predictive Factors of Oncologic Outcomes

The impact of all the factors considered on the oncologic outcomes in uni- and multivariate analyses are presented in Table 3.

The negative independent impact of salvage TL on oncologic outcomes (OS, CSS and RFS) was confirmed in multivariate analysis. Hypopharyngeal tumor site, ASA score ≥ 3, N-stage ≥ 2a and positive surgical margins were the main other predictors of poor prognosis.

ROC curves with their AUC for the three Cox regression models (OS, CSS and RFS) are presented in Figure 4. Of note, an AUC > 0.80 were obtained for the three ROC curves indicating a good predictive ability for the three Cox regression models.

## 4. Discussion

To our knowledge, the present study is the first to demonstrate the prognostic role of salvage TL compared to primary TL for patients with a locally advanced laryngeal or hypopharyngeal carcinoma (T3 or T4) at initial presentation. Indeed, previous studies on this subject had included in the salvage TL group a variety of tumors, especially second primary carcinomas developed in a previously irradiated neck and recurrences of early stage tumors (T1 or T2) initially treated by partial surgery (endoscopic or external) or by RT [16,17]. Obviously, this inclusion bias leads to an overestimation of patient survival rates in the salvage TL group. However, in terms of oncologic results, it is the comparison of primary vs. salvage TL for patients who are all initially eligible for TL that is meaningful and raises the question of the proper selection of patients for LP. This negative impact of salvage surgery was independently confirmed in multivariate analysis on the three survival parameters studied (OS, CSS and RFS).

This multivariate analysis was even more necessary as the two groups compared (primary vs. salvage TL) were not similar in terms of baseline clinical characteristics and, particularly, in terms of tumor site distribution (larynx vs. hypopharynx). Indeed, hypopharyngeal carcinomas are known to be more aggressive than laryngeal carcinomas, with notably higher submucosal, lymphatic and metastatic spread [20,21]. Thus, it could be argued that the difference in survival between primary and salvage TL was related to the over-representation of hypopharyngeal carcinomas in the salvage TL group compared to the primary TL group. However, the multivariate analysis confirmed the independent negative impact on survival outcomes of both salvage surgery and hypopharyngeal tumor site.

Moreover, the subgroup analysis according to the tumor site clearly validated the negative impact of salvage surgery on oncologic outcomes in the subgroup of patients with laryngeal carcinomas. For hypopharyngeal carcinomas, the negative impact of salvage surgery on patient survival also appeared to be present with worse crude survival rates than the primary TL group but with a statistical significance threshold that was not reached due to the limited number of patients. The differences observed in terms of survival were indeed major with survival rates almost halved between primary and salvage TL. Thus, the 5-year OS rates were 64% vs. 38% for laryngeal carcinomas and 34% vs. 19% for hypopharyngeal carcinomas in the primary and salvage TL groups, respectively.

The large LP studies performed in the 1990s and early 2000s were based on very strict patient selection [1,2,3,4,5,6,7]. For studies involving ICT, the assessment of tumor response to ICT based on both objective tumor response (tumor volume) and laryngeal remobilization, was also a very strict condition for continuing the LP protocol [3,4,5,7]. In addition, several patients included in these LP studies had tumors of T-stage ≤ 2, without laryngeal fixation, but with lymph node extension (explaining the overall stage III–IV), for which a conservative surgery was an option [1]. Thus, in the VA study, 10% of patients had a T stage ≤ 2 tumor and 43% of patients did not have laryngeal fixation (a major criterion contraindicating partial surgery) [4]. The Radiation Therapy Oncology Group (RTOG) study conducted by Forastiere et al., 2003 [6] included 20% T2 tumors and 44% mobile larynx [6].

At the opposite, patients with a T4-stage tumor invading the thyroid cartilage are not good candidates for LP and should theoretically be referred to radical surgery [3,8]. However, for more than 20 years, in daily practice, the selection of patients included in LP protocols has not been as rigorous and a certain number of patients have not been redirected to TL or were even refused this procedure despite the absence of significant response to ICT [1,3,8]. This explains why the results of LP protocols outside of clinical trials are not as promising as those reported in the original studies. At the same time, the possibilities for surgical salvage of recurrent tumor after a LP program are not as good in clinical practice as they were in the early studies. An update of the RTOG 91-11 study focusing on the quarter of patients who required a salvage TL showed a very favorable two-year OS rate of 72% (compared to 38% in our study for the larynx subgroup) [11]. This result may appear very similar to what can be achieved after a primary TL, leading to the erroneous, yet widespread consideration that there is always time to perform salvage TL after an LP failure and that it does not compromise survival. However, a critical analysis of patient selection shows that many of the patients who underwent salvage TL in the RTOG 91-11 study did not correspond to the indications for primary TL at initial diagnosis [6,11]. Although overall tumor stage is usually well correlated with patient prognosis, it is important to accurately describe T- and N-stage separately, to better define patient selection criteria. For this reason, in the present study, we chose to independently analyze the impact of T- and N-stage on survival outcomes.

Several retrospective studies have more recently analyzed a large series of salvage TL to assess the results of this surgery without comparison to a primary TL group [12,13,14,15]. Sandulache et al. reported a 5-year OS rate of 57% in 218 patients undergoing salvage surgery for laryngeal carcinoma, but most of them presented a stage I-II tumor at diagnosis [12]. There is considerable variability in the oncological results of salvage TL in the literature, the populations of the different studies being rarely superimposable [11,12,13,14,15,16,17,18,19]. To increase the volume of patients for statistical analysis, many authors included patients with T1-T2 tumors relapsing after RT or conservative surgery [11,12,13,14,15,16,17,18,19]. In the present study, we chose to exclude these patients, to keep only those presenting with a stage T ≥ 3 tumor at initial diagnosis for whom a primary TL could be discussed.

In another study of 244 patients undergoing salvage TL for laryngeal carcinoma, Birkeland et al. found a 5-year OS of 49% [18]. In this study, which did not yet include only patients with an initial stage T ≥ 3, the authors also showed that a high comorbidity index was the main factor of poor prognosis [18]. This negative role of the level of comorbidity on OS is usually found in surgical oncology [22]. In this regard, in multivariate analysis, we also showed the negative prognostic role of an ASA score ≥ 3 on both OS and DFS. This highlights the importance of selecting patients who are candidates for salvage surgery, whose prognosis is already poor and could also be threatened by uncontrolled comorbidities.

Few studies on salvage TL have focused on hypopharyngeal carcinomas. It is likely that in the setting of recurrent locally advanced tumors, it can sometimes be challenging to distinguish hypopharyngeal carcinomas from those arising in the supraglottis. In our series, the hypopharyngeal tumor site was an independent factor of poor prognosis which has also been reported in several studies [19,22,23]. Detailed analysis of the survival curves (Figure 3a–c) of the present study also showed that the impact of tumor site appeared to be greater in the primary TL group (well-separated larynx vs. hypopharynx curves) than in the salvage group. The salvage TL subgroup for hypopharyngeal carcinoma had the poorest prognosis with particularly poor survival rates (3- and 5-year OS of 25% and 19%, respectively). Even in LP trials, long-term survival rates for patients with hypopharyngeal carcinoma are particularly poor [5,7]. For example, in the EORTC LP study (trial 24891), patients in the ICT group had 5- and 10-year OS rates of 21.9% and 8.7%, respectively [5]. In a multicenter study of 21 patients who underwent salvage circular TL for recurrent hypopharyngeal carcinoma, Fakhry et al. showed a 5-year OS rate of 16% [24]. For hypopharyngeal carcinoma patients, the indication of a salvage TL must therefore be carefully weighed, especially since it is accompanied by a high morbidity and mortality [19,24]. Rigorous patient information is crucial here, as well as considering the patient’s overall situation (tumor extension, previous treatment, general health status, comorbidities, psychosocial status, etc.) [25]. The appropriate selection of patients for salvage surgery is indeed one of the most difficult decisions in head and neck oncology and can only be made in the context of a multidisciplinary discussion integrating the patient’s preferences.

Among the other poor prognostic factors identified in the present study, the role of N stage ≥ 2a should be noted, with a high statistically significant negative impact on the three survival end-points (OS, CSS and RFS). These results are consistent with the literature [26,27,28,29]. In a study of 316 patients with all stages of laryngeal carcinoma, Haapaniemi et al. reported a recurrence rate of 22% and showed that the initial presence of lymph node metastases was an independent predictor of recurrence [26]. In another study of 105 patients with recurrent head and neck carcinoma, Agrawal et al. showed that in patients who underwent salvage surgery, early initial tumor stage and isolated local recurrence without lymph node involvement were associated with a better prognosis [27]. Lymphatic spread is therefore a major factor to consider in therapeutic decision making, particularly before salvage TL, given the poor prognosis of the patients. The probability of achieving free surgical margins is also an important element to consider since our study showed, as is usual in oncologic surgery, the independent prognostic impact of the surgical margin status [22,23,24].

The increase in mortality of laryngeal carcinomas observed in the USA over the last 30 years could be interpreted in the light of the results of our study [10]. A policy of LP “at all cost” could have led to a decrease in patient survival [1,2,3,30]. Obviously, the present study is not intended to discuss the interest of LP and the benefits it has brought to patients for many years, particularly in terms of quality of life. Results of LP studies have indeed allowed many patients to preserve a functional larynx without compromising their survival [3]. However, the present study underlines the importance of an accurate selection of patients who are candidates for LP, knowing that the negative prognostic impact of tumor recurrence after a LP program will not be reversed by salvage surgery.

The main limitations of our study are its retrospective and single-center study design. The integration of data from other referral centers would increase the number of patients in the different subgroups, especially for salvage TL, and thus increase the power of our study. Analyzing together patients with laryngeal and hypopharyngeal carcinoma who have a very different prognosis may represent a weakness of our study. Nevertheless, this enabled us to assess the impact of the primary tumor site (hypopharynx vs. larynx) on patient oncologic outcomes and to evaluate separately the prognostic effect of the type of surgery (salvage vs. primary TL) in these two subgroups of patients. Moreover, it should be noted that the salvage TL group did not include all the failures of LP since some patients with recurrent tumors after an LP program are not candidates for salvage surgery (unresectable tumor, metastatic disease, poor general health status, etc.). The prognosis of this type of patients is certainly even worse than that of patients selected for salvage TL. Furthermore, our study did not address the impact of salvage surgery on postoperative complications, functional results and patient quality of life in comparison with primary TL [31].

## 5. Conclusions

Salvage TL is associated with significantly worse survival rates than primary TL. This adverse impact of salvage TL on patient survival was confirmed in multivariate analysis taking into account other relevant clinical factors such as patient comorbidity level, tumor site, N-stage and surgical margin status that were identified as the main other independent prognostic factors. These factors should be considered in the therapeutic decision-making, especially in the setting of salvage TL, given the poor prognosis of these patients.

The decision of enrolling a patient in an LP program must be taken by an MTB. There is great variability between centers regarding the applicability of clinical trial results on LP approaches. To translate the results of these trials into clinical practice, it is essential to strictly follow the study protocols particularly regarding eligibility criteria. Patients with T4a laryngeal or hypopharyngeal carcinoma, as well as those with poor pretreatment laryngoesophageal function should be referred to immediate TL.

Comparison of salvage vs. primary TL in terms of functional results (swallowing and voice outcomes) has to be explored further in future research. There is also a need to compare the two validated approaches for LP (i.e., TPF ICT followed by RT in responders vs. concurrent cisplatin-based CRT) using a modern composite primary end-point such as laryngoesophageal dysfunction-free survival. This is the objective of the ongoing French phase III trial (GORTEC 2014-03-SALTORL, clinicaltrials.gov NCT03340896).

## Figures and Tables

**Figure 1 jcm-12-01305-f001:**
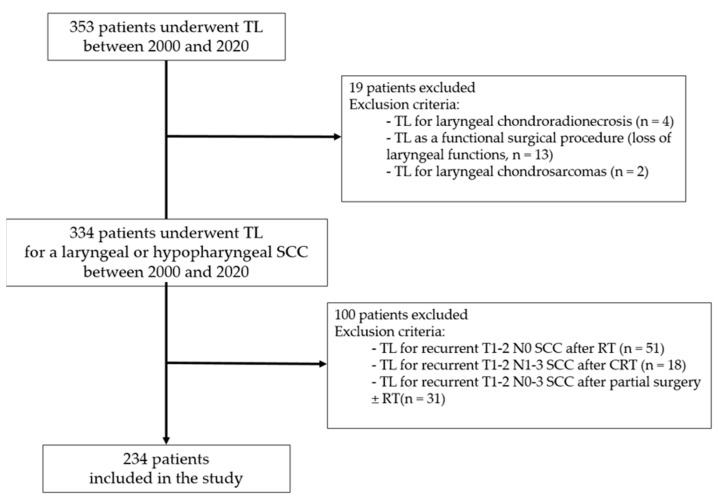
Flowchart of patient selection. TL: total laryngectomy; SCC: squamous cell carcinoma; RT: radiation therapy; CRT: chemoradiation therapy.

**Figure 2 jcm-12-01305-f002:**
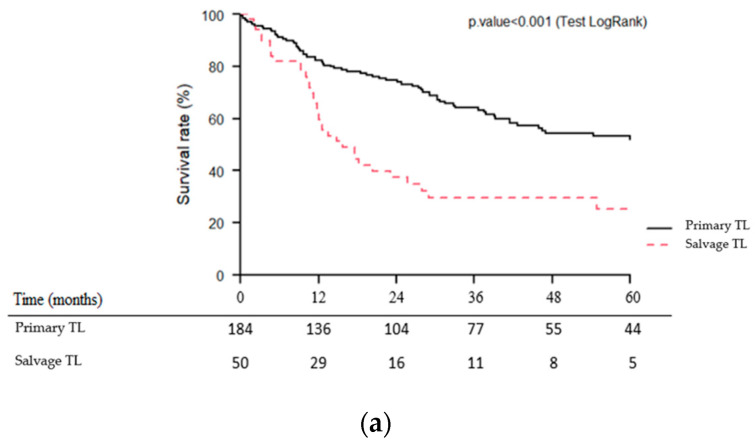
Kaplan–Meier overall (**a**), cause-specific (**b**) and recurrence-free (**c**) survival curves for patients undergoing primary vs. salvage total laryngectomy (TL).

**Figure 3 jcm-12-01305-f003:**
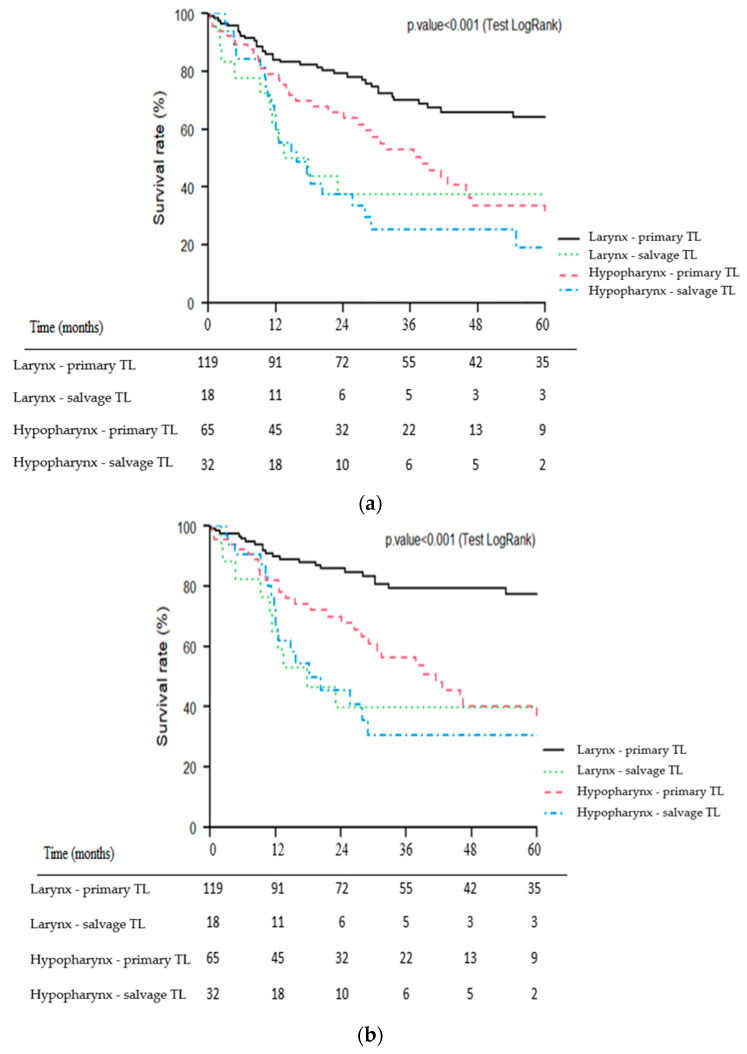
Kaplan–Meier overall (**a**), cause-specific (**b**) and recurrence-free (**c**) survival curves for patients undergoing primary vs. salvage total laryngectomy (TL) according to primary tumor site (larynx vs. hypopharynx).

**Figure 4 jcm-12-01305-f004:**
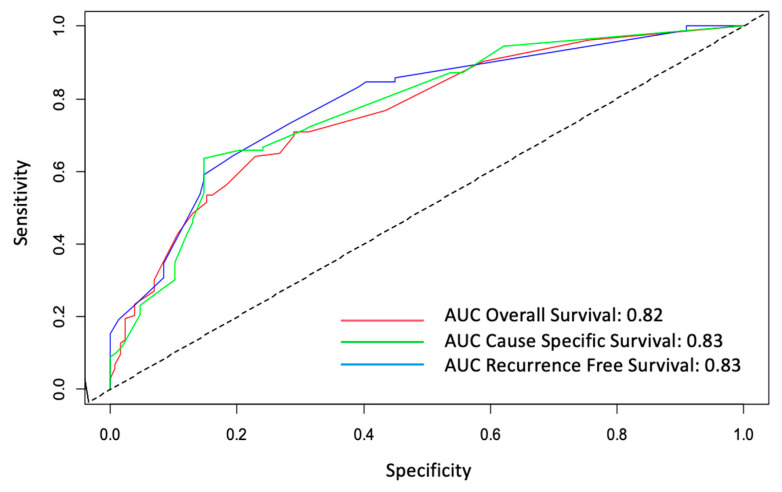
ROC curves for overall (OS), cause-specific (CSS) and recurrence-free survivals (RFS). AUC: area under the curve.

**Table 1 jcm-12-01305-t001:** Patients’ clinical characteristics according to the type of surgery.

Characteristics	All patients n = 234 (%)	Primary TL n = 184 (%)	Salvage TL n = 50 (%)	*p*
Gender: male/female	209 (89)/25 (11)	168 (90)/16 (10)	43 (86)/7 (14)	0.55
Age > 70 years	94 (40)	80 (43)	14 (28)	0.06
ASA score ≥ 3	114 (49)	90 (49)	24 (48)	0.43
Tumor site: larynx/hypopharynx	137 (58)/97 (42)	119 (65)/65 (35)	18 (36)/32 (64)	<0.0001
T-stage ^a^: T3/T4	106 (45)/128 (55)	66 (36)/118 (64)	40 (80)/10 (20)	<0.0001
N-stage ^a^: </≥2a	161 (69)/73 (31)	133 (72)/51 (28)	28 (56)/22 (44)	0.02
Neck dissection	206 (88)	170 (92)	36 (72)	0.0001
Pedicled flap	91 (39)	50 (27)	41 (82)	<0.0001
Free flap	40 (17)	26 (14)	14 (28)	0.03
TL with circular pharyngectomy	41 (17)	26 (14)	15 (30)	0.016
Tracheotomy prior to TL	54 (23)	44 (24)	10 (20)	0.69
Lymph node extracapsular spread	69 (30)	55 (29)	14 (28)	0.93
Positive surgical margins	30 (13)	18 (10)	12 (24)	0.015

ASA: American Society of Anesthesiologists, TL: total laryngectomy, *p*: *p*-values using Chi-2 tests. ^a^ TNM-stage at initial presentation before primary treatment.

**Table 2 jcm-12-01305-t002:** Oncologic outcomes according to type of surgery and tumor site.

Oncologic Outcomes	2-Year Rate (%)			3-Year Rate (%)			5-Year Rate (%)		
	OS	CSS	RFS	OS	CSS	RFS	OS	CSS	RFS
Larynx − Primary TL − Salvage TL									
	79	86	71	70	79	63	64	77	58
	38	40	33	38	40	33	38	40	33
Hypopharynx − Primary TL − Salvage TL									
	66	70	44	53	56	38	34	40	21
	37	45	20	25	31	17	19	31	8

OS: overall survival, CSS: cause-specific survival, RFS: recurrence-free survival, TL: total laryngectomy.

**Table 3 jcm-12-01305-t003:** Predictors of oncologic outcomes in univariate and multivariate analyses.

Predictive Factors	OS	CSS	RFS
	p(UA)/(MA)	p(UA)/(MA)	p(UA)/(MA)
Age (< vs. >70 years)	0.54/-	0.81/-	0.70/-
Gender (male vs. female)	0.009/0.11	0.12/-	0.11/-
ASA Score (< vs. ≥3)	<0.001/0.002 HR: 1.83 [1.25–2.69]	0.04/0.24	<0.001/0.004 HR: 1.66 [1.17–2.34]
Tumor site (L vs. H)	<0.0001/0.12	<0.0001/0.02 HR: 1.82 [1.13–2.92]	<0.0001/0.024 HR: 1.53 [1.06–2.22]
Type of surgery (primary vs. salvage TL)	<0.0001/0.0008 HR: 2.32 [1.53–3.53]	<0.0001/<0.0001 HR: 3.27 [2.02–5.28]	<0.0001/< 0.0001 HR: 2.55 [1.72–3.80]
T-stage (T3 vs. T4)	0.06/0.46	0.02/0.03 HR: 3.72 [1.15–12.1]	0.26/-
N-stage (< vs. ≥2a)	<0.0001/0.0002 HR: 2.42 [1.59–3.68]	<0.0001/<0.0001 HR: 3.06 [1.92–4.87]	<0.0001/<0.0001 HR: 2.57 [1.74–3.79]
Tracheotomy prior to TL	0.78/-	0.66/-	0.86/-
Positive surgical margins	<0.0001/0.03 HR: 1.67 [1.02–2.73]	<0.0001/0.03 HR: 1.77 [1.05–2.99]	<0.001/0.07

ASA: American Society of Anesthesiologists, L: larynx, H: hypopharynx, TL: total laryngectomy, OS: overall survival, CSS: cause-specific survival, RFS: recurrence-free survival. p(UA)/(MA): *p*-values in univariate analysis (Log-Rank tests) and multivariate analysis (Cox regression models). Significant *p*-values are underscored and followed by their corresponding Hazard Ratios (HR) with their 95% confidence intervals.

## Data Availability

The data presented in this study are available upon request from the corresponding author. The data are not publicly available due to privacy.

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
