# Peer review of "Salvage vs. Primary Total Laryngectomy in Patients with Locally Advanced Laryngeal or Hypopharyngeal Carcinoma: Oncologic Outcomes and Their Predictive Factors"

_jcm, 2023, doi:10.3390/jcm12041305_

Round 1
Reviewer 1 Report
Attached in word

Reviewer 2 Report
1. What is the main question addressed by the research? Salvage vs primary total laryngectomy in patients with locally advanced laryngeal or hypopharyngeal cancer: oncologic outcomes and their predictive factors. 2. Do you consider the topic original or relevant in the field? Does it address a specific gap in the field? Accepted. However, it can be improved. 3. What does it add to the subject area compared with other published material? Yes. 4. What specific improvements should the authors consider regarding the methodology? What further controls should be considered? Please check my comments. 5. Are the conclusions consistent with the evidence and arguments presented and do they address the main question posed? Yes. 6. Are the references appropriate? Please check my comments.
Further Comments:
1. Abstract - too many keywords. 5 are appropriate.
2. Introduction - can be improved. The previous works / past researches can also be added.
3. Materials & Methods - A flowchart of methodology can be added to improve the presentation. It also can provide a better understanding of the proposed method.
4. Results - It is recommended to add a ROC curve can provide a better picture of the research.
5. Conclusion - Future work and recommendation can be added.
6. References - 68% of references is more than 5 years. The latest papers must be added into the manuscript. Must be improved.
Reviewer 3 Report
Thank you for the opportunity to review your manuscript about “Salvage vs primary total laryngectomy in patients with locally advanced laryngeal or hypopharyngeal cancer: oncologic outcomes and their predictive factors”. Overall, I believe this study has potential scientific utility. Although the data is convincing, the novelty of this study is limited. I point out some problems of this study.
Major Comments:
1) Authors used a LP program. Authors should describe a LP program in details in this study.
2) There are some differences in important clinical characteristics between
there two groups. Did these affect results of this study ?
3) Hypopharyngeal and laryngeal cancers are analyzed simultaneously. Hypopharyngeal and laryngeal cancers have different prognoses and surgical procedures. I believe that discussing them together makes it difficult to interpret the results. It is easier to understand this study if authors focus on either laryngeal cancer or hypopharyngeal cancer.
Round 2
Reviewer 1 Report
I am delighted with your changes. Now your work presents as a solid and well-thought-out study that will certainly improve the research on laryngeal and hypopharyngeal cancer.
My only remaining concern is your conclusions the first paragraph fits your study aims perfectly, but the second and third paragraphs read more like a discussion or practical recommendations. I will leave this decision whether to move/remove/leave it as it is for your team.
I sincerely hope to see more publications from your team.
Reviewer 2 Report
All comments have been addressed.
Reviewer 3 Report
Thank you for the opportunity to review your revised manuscript. I am satisfied with the revisions and explanation that have been made by authors.